# Compartmentalized Innate Immune Response of Human Fetal Membranes against *Escherichia coli* Choriodecidual Infection

**DOI:** 10.3390/ijms23062994

**Published:** 2022-03-10

**Authors:** Andrea Olmos-Ortiz, Mayra Hernández-Pérez, Pilar Flores-Espinosa, Gabriela Sedano, Addy Cecilia Helguera-Repetto, Óscar Villavicencio-Carrisoza, María Yolotzin Valdespino-Vazquez, Arturo Flores-Pliego, Claudine Irles, Bruno Rivas-Santiago, Elsa Romelia Moreno-Verduzco, Lorenza Díaz, Verónica Zaga-Clavellina

**Affiliations:** 1Departamento de Inmunobioquímica, Instituto Nacional de Perinatología (INPer), Mexico City 11000, Mexico; nut.aolmos@gmail.com (A.O.-O.); mayrahp10@gmail.com (M.H.-P.); m.pilar.flores.e@gmail.com (P.F.-E.); gasego_2012@hotmail.com (G.S.); addy.helguera@inper.gob.mx (A.C.H.-R.); cuauqbp@gmail.com (Ó.V.-C.); arturo_fpliego@yahoo.com.mx (A.F.-P.); 2Departamento de Anatomía Patológica, INPer, Mexico City 11000, Mexico; yolotzv@gmail.com; 3Departamento de Fisiología y Desarrollo Celular, INPer, Mexico City 11000, Mexico; cloirles@gmail.com; 4Unidad de Investigación Médica IMSS Zacatecas, Zacatecas 98000, Mexico; rondo_vm@yahoo.com; 5Subdirección de Auxiliares de Diagnóstico, INPer, Mexico City 11000, Mexico; elsamover@yahoo.com; 6Departamento de Biología de la Reproducción, Instituto Nacional de Ciencias Médicas y Nutrición Salvador Zubirán, Mexico City 14080, Mexico; lorenzadiaz@gmail.com

**Keywords:** genitourinary infection, innate defense, innate immunity, antimicrobial peptides, beta defensins, alpha defensins, maternal–fetal interface, collagen degradation

## Abstract

An infectious process into the uterine cavity represents a major endangered condition that compromises the immune privilege of the maternal–fetal unit and increases the risk for preterm birth (PTB) and premature rupture of membranes (PROM). Fetal membranes are active secretors of antimicrobial peptides (AMP), which limit bacterial growth, such as *Escherichia coli*. Nevertheless, the antibacterial responses displayed by chorioamniotic membranes against a choriodecidual *E. coli* infection have been briefly studied. The objective of this research was to characterize the profile of synthesis, activity, and spatial distribution of a broad panel of AMPs produced by fetal membranes in response to *E. coli* choriodecidual infection. Term human chorioamniotic membranes were mounted in a two independent compartment model in which the choriodecidual region was infected with live *E. coli* (1 × 10^5^ CFU/mL). Amnion and choriodecidual AMP tissue levels and TNF-α and IL-1β secretion were measured by the enzyme-linked immunosorbent assay. The passage of bacterium through fetal membranes and their effect on structural continuity was followed for 24 h. Our results showed that *E. coli* infection caused a progressive mechanical disruption of the chorioamniotic membranes and an activated inflammatory environment. After the challenge, the amnion quickly (2–4 h) induced production of human beta defensins (HBD)-1, HBD-2, and LL-37. Afterwards (8–24 h), the amnion significantly produced HBD-1, HBD-2, HNP-1-3, S100A7, sPLA2, and elafin, whereas the choriodecidua induced LL-37 synthesis. Therefore, we noticed a temporal- and tissue-specific pattern regulation of the synthesis of AMPs by infected fetal membranes. However, fetal membranes were not able to contain the collagen degradation or the bacterial growth and migration despite the battery of produced AMPs, which deeply increases the risk for PTB and PROM. The mixture of recombinant HBDs at low concentrations resulted in increased bactericidal activity compared to each HBD alone in vitro, encouraging further research to study AMP combinations that may offer synergy to control drug-resistant infections in the perinatal period.

## 1. Introduction

According to the World Health Organization, preterm birth (PTB) is defined as childbirth occurring at less than 37 completed weeks or 259 days of gestation from the first date of the last menstrual period. This condition is considered a major public health problem. In its 2019 report, the WHO estimated the worldwide rate of PTB in 2014 was 10.6%, which translated into approximately 14.84 million live preterm births [1]. Children who are born prematurely have higher rates of cerebral palsy, sensory deficits, learning disabilities, and respiratory illnesses compared with children born at term. The morbidity associated with PTB often extends to later life, resulting in enormous physical, psychological, and economic costs and an increased mortality rate during adulthood [2].

One of the main causal factors associated with PTB is an ascending intrauterine infection. A microorganism that is consistently associated with serious maternal and perinatal complications is *Escherichia coli* (*E. coli*), which is a versatile Gram-negative microorganism that is a well-known commensal of the normal intestinal microbiome. During pregnancy, *E. coli* invasion of the urinary and cervicovaginal tracts represents a high-risk condition that has been consistently associated with adverse pregnancy outcomes, including early and late miscarriage [3], preterm premature rupture of membranes (pPROM) [4], PTB [5], maternal or neonatal sepsis [6,7], stillbirth [8], and chorioamnionitis. The bacterial etiology of clinical chorioamnionitis is usually polymicrobial, with the most common being *Ureaplasma urealyticum* and *Mycoplasma hominis* [9]. However, isolation of *E. coli* in infected mothers has been increasing in the last decades. *E. coli* was detected in up to 12.5% of chorioamnionitis cases during the 1980s and 1990s [10,11]; whereas in recent histological analysis, a considerable proportion of chorioamnionitis cases are of *E. coli* origin (ranging from 25–43%) [12,13].

During an ascending bacterial infection, the chorioamniotic membranes are the ultimate barrier that the pathogen must cross before reaching the amniotic cavity and eventually the fetus. Therefore, in an adverse scenario such as this, the immune competencies of these membranes, including operative mechanisms to prevent bacterial proliferation, can represent the difference between the continuity and the interruption of gestation. There is robust evidence that supports the notion that fetal membranes are active secretors of diverse molecules of innate immunity, including natural antimicrobial peptides (AMP), which provide broad-spectrum protection against bacteria, yeast, and some viruses [14,15,16]. Furthermore, there is evidence supporting that the amnion and choriodecidua exert differential immune competencies, which allow them to respond selectively against different microbial insults [17].

AMP are small, cationic, amphiphilic, microbicidal peptides that play a key role in both innate and adaptive immunity, mainly produced by epithelial surfaces and immune cells [14]. They represent a major defense mechanism of the female reproductive tract before and during pregnancy as the first-line defense of the innate immune system against a pathogen’s invasion [14,18], and they are of special interest in PTB prevention [19]. Important and broadly studied AMP include human beta defensins (HBDs), which play a key role in the mechanisms involved in the protection of maternal and fetal tissues from uterine infections. HBDs are profusely expressed by the decidua, the amnion epithelium, and placental and chorion trophoblasts [20,21,22]. Another subclass of defensins is the alpha defensins, which include the human neutrophil peptides (HNP)-1-3, also named human neutrophil defensins. During pregnancy, they are expressed in the cervical mucus plug [23], placenta [24], chorioamniotic membranes [25], and in the amniotic fluid [26]. Interestingly, in vitro assays showed an important antimicrobial activity of HNP-1-3 against *E. coli* [27].

Besides defensins, there are other proteins and peptides belonging to the AMP family, including elafin, ribonucleases, phospholipases, S100A7, and cathelicidin, among others. Elafin is a 9.9 kDa peptide that is expressed in the non-pregnant endometrium, cervix, and vagina, while during pregnancy it is present in the amnion, chorion, placenta, decidua, and the cervical glandular epithelium [21,28]. Among the ribonuclease A superfamily, RNase-7 and RNase-8 present a broad spectrum of antimicrobial activity against pathogenic bacteria and fungi [29,30]. In addition, phospholipase A2s not only participate in the regulation of phospholipid metabolism at the cellular membrane but are also potent AMPs related to the control of infectious diseases. Among these proteins, type-IIA secretory phospholipase A2 (sPLA2-IIA) displays the highest bactericidal activity [31]. Psoriasin or S100A7 belongs to the S100 family related to cell cycle progression and cellular differentiation. Its expression increases in response to inflammatory stimuli, and it exhibits antimicrobial activities against bacteria and fungi, although inducing immunomodulatory activities [32]. Human cathelicidin, also known as LL-37, stands as an essential cationic component of the host innate immune response, with strong microbicidal activity against a wide range of microorganisms [33]. Its microbicidal properties were first observed in neutrophils, but it is known that many tissues can produce it, including the vagina, cervix [34], and placental trophoblast [35].

In a previous study, we showed that the chorioamniotic membranes differentially secreted HBD-1, HBD-2, and HBD-3 within 24 h of *E. coli* insult, depending on which compartment the infection was applied [36]; however, to date, there is still little information about bacterial migration pattern and the temporal- and tissue-dependent regulation of the innate defense AMP synthesis deployed against this pathogen in the amnion (AMN) and choriodecidua (CHD) layers, despite the strong evidence linking *E. coli* colonization in the choriodecidua and many adverse pregnancy and birth outcomes [3,4,5,6,7,8]. Therefore, the main objective of the present research was to widen our previous study by characterizing the tissue content of a robust panel of different AMP (HBD-1-4, HNP-1-3, elafin, RNase-7, RNase-8, sPLA2, S100A7, and LL-37) in a time-lapse manner in a model of choriodecidual infection with live *E. coli*, which allowed us to reproduce in vitro a pathological condition, such as an ascending intrauterine infection. Additionally, we wanted to analyze the growth and migration patterns of *E. coli* and its effect on the mechanical integrity of the choriodecidual and the amniotic compartments. Finally, we probed in vitro the microbicidal functional activity of recombinant HBDs.

## 2. Results

### 2.1. Bacterial Migration through Chorioamniotic Membranes and Their Associated Structural Damage

The histological examination of *E. coli*-infected membranes was analyzed 2, 4, 8, and 24 h after tissue inoculation to evaluate the mechanical damage and bacteria migratory capacity through the CHD and AMN layers. The control tissues preserved their structural integrity (Figure 1A,L) and had no evidence of bacterial bodies (Figure 1F). At 2 h post-infection, slight extracellular matrix degradation was noticed mostly in the compact layer (shown in Figure 1M), while bacterial bodies were visible in the CHD and AMN (Figure 1G). At 4 h post-infection, *E. coli* proliferation was detected in Gram-stained sections, where bacterial spreading and invasion through the choriodecidual bed and into the amnion could be observed (Figure 1H). Disruption of amnion mechanical integrity was evident, particularly within the intermediate and compact layers (Figure 1C,N). At 8 h post-infection, bacterial presence in the distal AMN, close to amniotic epithelia, was more evident (shown in Figure 1I) while it disrupted the mechanical integrity of the amnion (Figure 1D,O). Finally, at 24 h, the CHD and AMN layers were severely injured by *E. coli*, which resulted in a total mechanical rupture of the extracellular matrix of the AMN (Figure 1E,P) as well as bacterial accumulation in the CHD (Figure 1J) and amniotic epithelia (Figure 1K).

### 2.2. Inflammatory Milieu Induced by E. coli Infection in Chorioamniotic Membranes

*E. coli* infection was corroborated not only at the microscopic level but also by the inflammatory milieu developed in response to these bacteria. Choriodecidual infection with *E. coli* resulted in a progressive increase in TNF-α secretion to culture media in both compartments (Figure 2). As depicted, the CHD secreted significantly more TNF-α at 4 and 8 h post-infection in comparison to the non-infected membranes, while the AMN significantly induced TNF-α only at 8 h post-infection, which coincides with the disrupted integrity of the chorioamniotic membranes observed at this time-point (Figure 1D,O). IL-1β was also measured in the culture medium; however, it was undetectable until 24 h post-infection (9116 ± 3930 pg/g of tissue in the CHD and 6150 ± 4830 pg/g of tissue in the AMN, *p* < 0.05 vs. control values, which were considered as zero since they were below the detection limits of the assay).

### 2.3. Transmembrane Migration of Bacteria from Choriodecidua to Amnion Media Culture

Because of the observed mechanical disruption of the chorioamniotic membranes, we expected to detect the presence of bacteria in the CHD medium culture but also on the AMN side. As shown in Figure 3, at 2 and 4 h post-infection, the density of bacteria initially inoculated (1 × 10^5^ colony-forming units (CFU)/mL) showed no statistical variation and persisted only in the choriodecidual compartment. However, at 8 h post-infection there was a significant increase in the CFU/mL of the bacteria in the CHD culture medium compared to the inoculum. Moreover, an important and significant entrance of bacteria was observed in the AMN culture medium, which may reflect the disruption of the mechanical barrier previously observed at this time point (Figure 1D,O). At 24 h post-infection, bacteria proliferated significantly in both the AMN and CHD, up to 1 × 10^10^ CFU/mL, with no difference between these compartments’ media, which corroborated a total disruption of the chorioamniotic membrane’s physical barrier (Figure 1E).

### 2.4. Antimicrobial Peptides Synthesis Profile in Response to E. coli Choriodecidual Infection

We followed the kinetic profiles of diverse AMP that were synthesized in response to *E. coli* choriodecidual infection. The first set of evaluated peptides included HBDs (Figure 4). HBD-1 was basally secreted by both the CHD and AMN with no differences between these compartments at time zero. However, there was a significant increase in HBD-1 synthesis by the AMN at 2, 4, 8, and 24 h post-infection in comparison to the control cultures. In contrast, HBD-1 concentration in the CHD did not change over time during the follow-up (Figure 4A). Regarding HBD-2, the AMN showed a significantly higher production of this peptide when compared to the CHD at all time points. Whereas HBD-2 synthesis by the CHD was not modified by a bacterial infection, in the AMN there was a significant increase in the production of this peptide compared to the control cultures at 2, 4, and 8 h, returning to basal levels at 24 h (Figure 4B).

On the other hand, there was a constitutive basal expression of HBD-3 and HBD-4 throughout the incubation time, without changes after the bacterial challenge (Figure 4C,D). However, basal HBD-3 synthesis was significantly higher in the AMN than in the CHD.

The second peptide set evaluated includes alpha defensins and other peptides with microbicidal activity (Figure 5). HNP-1-3 were significantly induced by the bacterial challenge only at 24 h post-infection in the AMN compartment (Figure 5A). In the case of LL-37, its concentration was below the detection limits of the assay in the AMN culture media under the basal condition and, therefore, was significantly lower in comparison to the basal production of the CHD. Interestingly, *E. coli* infection robustly and significantly increased AMN LL-37 synthesis, starting 4 h post-infection. In contrast, the CHD compartment increased LL-37 production after 8 h and 24 h post-infection (Figure 5B). S100A7, elafin, and sPLA2 showed a similar profile; their production by the AMN compartment was significantly induced at 24 h post-infection in comparison to the controls, while the bacterial challenge did not modify these peptides synthesis in the CHD compartment (Figure 5C–E). Finally, we evaluated RNase-7 and -8 production by the fetal membranes. Both RNases were basally produced by the CHD and AMN but were not significantly modified in response to *E. coli* infection. RNase-7 basal content: 1704 ± 742 ng/mg protein in the AMN and 733 ± 417 ng/mg protein in the CHD. RNase-8 basal content: 35.7 ± 6.6 ng/mg protein in the AMN and 17.8 ± 2.3 ng/mg protein in the CHD).

### 2.5. In Vitro Microbicidal Activity of Recombinant Human Beta Defensins

Considering that only HBD-1 and HBD-2 were induced by *E. coli* in the AMN compartment, we wondered if HBD-3 and HBD-4 exerted a cooperative bactericidal effect with the other defensins due to their constitutive expression. As shown in Figure 6, at the 2000 ng dose tested, recombinant HBDs did not significantly prevent bacterial growth per se. However, the concomitant exposure of all defensins 1–4 (500 ng, each one) to the *E. coli* culture significantly decreased bacterial CFU/mL by one and two orders of magnitude at 4 h of incubation. The latter supports the hypothesis of a cooperative interaction between HBDs for acquiring greater bactericide effect (Figure 6).

## 3. Discussion

An infectious process into the uterine cavity represents a major jeopardizing condition that endangers the immune privilege of the maternal–fetal unit. Therefore, a rapid and efficient response of the human fetal membranes against any immunological/inflammatory challenge is critical and imperative for a successful pregnancy [37].

Our results document in a timeline the bacterial migratory capacity from the choriodecidual region to the amnion in a 24 h period. Bacteria attached to the CHD membrane actively proliferated during the first 4 h of culture, disrupted the collagen barrier, and compromised membrane integrity, allowing their dissemination and invasion of the amniotic region 24 h after the initial contact. Our results corroborate previous studies with scanning transmission electron microscopy demonstrating the ability of *E. coli* to attach and cross the chorioamniotic membranes from the maternal side to the amniotic side [38,39].

This study was mainly undertaken to characterize the compartmentalized profile of the synthesis and activity of a broad panel of AMP produced by human fetal membranes under basal conditions and upon a bacterial challenge. The present work described a temporal- and tissue-specific regulation of peptides from the innate response in chorioamniotic membranes infected with *E. coli.* Considering our results, we detected two temporal windows for the chorioamniotic synthesis of AMP in response to *E. coli* choriodecidual infection: an early response (2 to 4 h post-infection) and a late response (8 to 24 h post-bacterial challenge). In Figure 7, we summarize the data obtained in the present work as a schematic overview of the structural damage and the inflammatory milieu in response to *E. coli* infection as well as the synthesis of AMP by the CHD and AMN layers.

Our results showed that fetal membranes constitutively produced all the analyzed peptides and they remained within both sides, except for LL-37, which was absent in the AMN. In the fetal membranes, amniotic epithelial cells, which are in direct contact with the fetus, amniotic stromal mesenchymal cells embedded in the fibrous layer of connective tissue, and trophoblasts that are in contact with the maternal decidua are able to synthesize and secrete AMP in response to different stimuli [40,41].

We also observed that the AMN was the compartment that showed the highest tissue content of AMP when challenged with *E. coli*, even though inoculation was applied only in the choriodecidual side. As described by others, the epithelial nature of the amnion is probably involved in this over-response in comparison with the choriodecidua [14,15]. The present study complements a previous one that reported differential HBD secretions into the cell culture by the AMN and CHD after bacterial infection [36]. In that report, we found an important contribution of the CHD to defensin release into the medium, particularly HBD-2, while herein the AMN was the main contributor to tissue AMP content. Therefore, it seems that both intra-tissue as well as secreted AMP are key players that differentially contribute to chorioamniotic defense. In addition, the modulation of lymphocyte T activity and innate cell recruitment to the site of infection are important events that take place in fetal membranes and should not be discarded as potential defensive mechanisms against *E. coli* choriodecidual infection. Unfortunately, the nature of our in vitro model cannot demonstrate this, and it remains as an important question to resolve.

Herein we also studied the tissue damage that *E. coli* was inflicting while migrating from the CHD to the AMN. In this regard, we observed a fast and progressive mechanical disruption of the chorioamniotic membranes and an activated inflammatory environment, as shown by increased TNF-α and IL-1β secretion. TNF-α is the main inflammatory modulator induced after LPS stimulation of the choriodecidual region in both in vitro and in vivo models [42,43,44]. In contrast, IL-1β acts as part of a delayed tissue response to an immunological challenge [42], perpetuating the pro-inflammatory environment and the damage of membranes. Similar to previous reports, TNF-α is produced by the AMN and CHD, whereas IL-1β is secreted at detectable levels only by the CHD [45,46,47]. This supports the differential secretion profile that we observed in our results.

In an attempt to contain the bacterial infection, fetal membranes responded by increasing the synthesis of diverse AMP in a time- and location-dependent manner. Indeed, rapid production of HBD-1, HBD-2, and LL-37, mainly by the amnion, was observed as soon as 2 h post-infection. Later, additional AMP could be detected in this compartment, including HNP-1-3, S100A7, sPLA2, and elafin, whereas in the CHD, only cathelicidin was induced by *E. coli*. Although HBD-3, HBD-4, RNase-7, and RNase-8 were not significantly responsive to the infectious challenge, they altogether have a crucial role as part of the basal mechanism of defense by human fetal membranes and may also have a role to potentiate the antimicrobial activity of fetal membranes, considering our results for a cooperative microbicide interaction between HBDs.

Robust evidence indicates that inflammatory and infectious challenges stimulate the mRNA or protein levels of diverse AMP in the female reproductive tract. Clinical evidence shows that the concentration of HDB-1, -2, and -3 as well as HNP1-3 in amniotic fluid increases in pathological events, such as PTB, pPROM, PROM, microbial invasion of the amniotic cavity, intra-amniotic inflammation, and subclinical chorioamnionitis [48,49,50,51] Regarding this, several studies have shown that inflammatory stimuli, including cytokines, lipopolysaccharides, and bacteria, alter the expression and secretion of AMP in fetal membranes and placenta [21,52,53,54,55,56,57]. In particular, the role of IL-1β and TNF-α as in vitro inducers of the expression and secretion of HBD-1, HBD- 2, HBD-3, elafin, and LL37 has been consistently reported in placental trophoblast and amniotic cells [21,22,41]. Interestingly, the induction of AMP by TNF-α treatment in human fetal membranes results in a less noticeable induction than that exerted by the same dose of IL-1β [41]. In a previous report of our lab group, we demonstrated that TNF-α is the first cytokine released after LPS stimulation of fetal membranes. After this first wave, the proinflammatory state is maintained by a modest increase of IL-1β that reaches a peak at 48 h [42]. Therefore, it is plausible to hypothesize that the herein observed early induction of HBD-1, HBD-2, and LL-37 by AMN could be the result, at least in part, of the paracrine stimulation with TNF-α from the CHD, whereas IL-1β may regulate the delayed wave of AMP, which deserves to be further investigated.

Our results showed a significant antiproliferative effect of HBDs against *E. coli* when used altogether (HBD 1-4). Indeed, the combination of recombinant HBDs significantly inhibited bacterial growth as compared to each HBD alone. Our results agree with others showing a synergistic effect of HBDs and LL-37 against *E. coli* [58]. This may be explained considering that AMP often act on different bacterial targets and with diverse mechanisms of action: (i) AMP group and bind to the bacterial wall or membrane, form pores, keep them open, and prevent their repair; (ii) AMP can also bind to the bacterial wall, preventing peptidoglycan elongation; (iii) some AMP can be endocytosed or directly penetrate into the bacterial cytoplasm and affect several enzymes involved in vital processes (such as matrix metalloproteases, which are essential in microbial cell growth and homeostasis); (iv) some AMP present a high binding affinity for DNA and RNA and can act as inhibitors of nucleic acid biosynthesis, which disrupts bacterial replication and messenger transcription; (v) additionally, they can bind to several components of the translation machinery and inhibit protein translation and protein folding or lead to protein degradation; and vi) they inhibit the formation of biofilms [59,60]. In addition to the bacterial targets, AMP can also bind to host targets, and regulate their immune response: (i) AMP reduce the host inflammatory reaction caused by endotoxins; (ii) favor mast cell degranulation and histamine release, enhancing splenocyte and lymphocyte production of Th1 and Th2 cytokines; (iii) AMP have been shown to increase the production of chemokines released from immune and epithelial cells, favoring the chemoattraction of monocytes, neutrophils, mast cells, naïve T cells, CD8 T cells, immature dendritic cells, and endothelial cells to fight against infection [60,61]. Altogether, these complementary mechanisms (transient, permanent, or both), could result in bacterial killing.

Elsewhere, the synergy exerted by a mixture of AMP has been attributed to the differential stability of these peptides, the rapidness of formation of the bacterial pores, pores staying open for longer durations, and the perturbation of bacterial intracellular functions [62,63,64]. Interestingly, the effective antibacterial combination of recombinant HBDs used herein was achieved using a sub-lethal concentration, which is to say in the nanomolar range, as opposed to micromolar concentrations reported in previous assays [65,66]. To our knowledge, this is the first time that such a low HBD concentration resulted in a significant antibacterial activity against *E. coli*.

Despite the observed broad battery of produced AMP, the fetal membranes were unable to kill bacteria or contain the collagen degradation or the bacterial growth and migration, features associated with an increased risk of PROM and PTB. This weakened effect in our ex vivo model might be due to AMP degradation or the loss of bioactivity, most likely because of peptides’ lability to the proteases present in the cultured membranes. Besides that, in this approach, there was no presence of other immune cells that are habitually present in the decidua (such as uterine natural killers, macrophages, dendritic cells, and Treg cells), which can also contribute to the innate immune responses in vivo at the maternal–fetal interface [17]. Therefore, this observed quick (less than 24 h) degradation of the extracellular matrix and bacterial invasion of the AMN side could not represent the complete antimicrobial response deployed in vivo by fetal membranes against a cervico-vaginal *E. coli* infection; however, this approach is still the best fitted model to follow-up the chorioamniotic production of AMP as well as their mechanical properties in a bacterial challenge scenario.

## 4. Material and Methods

### 4.1. Ethics Statement

This study was approved by the Biosafety, Ethical and Research Committee from the Instituto Nacional de Perinatología (INPer) “Isidro Espinosa de los Reyes” and is registered under code number 212250-3210-21205-01-15. All methodological approaches were conducted according to the Belmont Report. Written informed consent, according to Declaration of Helsinki, was obtained voluntarily from each mother before caesarean section.

### 4.2. Biological Samples

Biological samples were collected from normoevolutive, uncomplicated, term (37–40 weeks) pregnant women, who attended their cesarean section in the Instituto Nacional de Perinatología (INPer) and gave their voluntary consent. Exclusion criteria included: vaginal delivery, cervicovaginal infection during the third trimester, diabetes mellitus, hypertension, obesity, or other metabolic diseases. All patients included in this study lived in Mexico City. They were Hispanic and of middle socioeconomic status. Clinical data from mothers and newborns are described in Table 1. We processed a total number of 13 chorioamniotic membranes: AMP synthesis and bacterial growth in fetal membrane culture media were measured in 6 of them, 5 were used for histological analysis, and 4–8 were used to evaluate *E. coli* antimicrobial activity.

### 4.3. Chorioamniotic Membrane Culture

In the present study, we used an experimental model that was originally validated and published by our team in which the fully functional fetal membranes are cultured in a two-compartment system that maintains viable AMN and CHD that are responsive and in paracrine communication [47].

Briefly, chorioamniotic membranes were manipulated under sterile conditions and were rinsed in sterile 0.9% sodium chloride solution to remove blood clots, and the whole membrane was visually inspected to ensure mechanical integrity. Fetal membranes were manually cut with a scalpel into ~3 × 3 cm and were placed over a synthetic membrane-depleted transwell device (Costar, New York, NY, USA). Chorioamniotic membranes were fixed in place using sterile silicone rubber rings. The CHD side was oriented in the upper chamber of the transwell, whereas the AMN side was oriented lower. This model allowed us to independently evaluate the secretions of the AMN and CHD. The transwell-assembled membranes were placed into a 12-well culture dish and maintained for 24 h in DMEM (Dulbecco’s modified Eagle’s medium)-supplemented culture medium (DMEM + 10% FBS + 1 mM sodium pyruvate + penicillin 100 U/mL, streptomycin 100 µg/mL, and amphotericin B 25 ng/mL) in each side of the chamber in a humidified incubator at 37 °C and 5% CO_2_–95% air. Additionally, small tissue sections were maintained for 3 days in selective bacterial media culture to ensure the absence of subclinical chorioamniotic infection by *E. coli*, *Staphylococcus aureus*, *Streptococcus agalactiae*, *Candida albicans*, *Gardnerella vaginalis*, *Neisseria gonorrhoeae*, *Lactobacillus* sp., *Klebsiella* sp., *Ureaplasma urealyticum,* or *Mycoplasma hominis*.

### 4.4. Choriodecidual Infection with Escherichia coli

The *E. coli* strain used herein was isolated from the blood of a neonate diagnosed with early onset sepsis and whose mother was diagnosed with PROM and chorioamnionitis, at the INPer. This strain was identified by means of a Vitek^®^ system and confirmed by its 16S rRNA sequencing. The *E. coli* genotype was determined by multiplex PCR (polymerase chain reaction) as the B2-phylogroup and the detected virulence genes were *PAI*, *papA*, *fimH*, *ibeA*, *fyuA*, *iutA* (*aerJ*), *hlyA,* and *traT*. Therefore, it was a pathogenic strain [67,68].

*E. coli* was grown in Luria–Bertani selective culture media at 37 °C overnight before the experimental treatments. After 24 h of culture, the transwell-assembled membranes were extensively washed with sterile 0.9% sodium chloride solution, and experimental treatments were carried out in DMEM-supplemented medium without antibiotics (DMEM antibiotic-free medium + 10% FBS + 1 mM sodium pyruvate) in triplicate or duplicate. Afterwards, CHD compartment was infected with 1 × 10^5^ CFU/mL of *E. coli*, which was corroborated in every experiment by counting the visible colonies in tryptic soy agar (TSA) plates as described in the next subheading. On the other hand, the AMN compartment was maintained exclusively with DMEM antibiotic-free medium (without inoculum). The non-infected control represented the chorioamniotic membranes incubated for 24 h with DMEM antibiotic-free medium in both compartments in the absence of bacteria. The infection rate of 1 × 10^5^ CFU/mL was chosen because this count in a urinalysis test represents a pathognomonic sign of urinary tract infection [69,70]. The kinetic time of infection was followed at 2, 4, 8, and 24 h (in triplicate). An aliquot of culture medium was frozen and was stored at −70 °C until cytokine quantification.

### 4.5. CFU Count and Bacterial Transmigration Assay in Cultured Fetal Membranes

At each time of the follow-up infection (2, 4, 8, or 24 h), an aliquot of culture medium was taken from the AMN and CHD chambers to quantify bacterial transmembrane passage. The count of viable bacteria in the culture medium was estimated by CFU counts in TSA plates. Briefly, serial 10-fold dilutions were seeded in triplicate into TSA plates. After overnight incubation at 37 °C, the plates with visible and countable CFU were chosen, and the corresponding dilution was registered. The final number of colonies in each treatment was estimated by the following Equation (1).
(1)$$\mathrm{CFU}/\mathrm{mL}=\frac{\left(\mathrm{No}.\;\mathrm{of}\;\mathrm{colonies}\right)\left(\mathrm{Dilution}\;\mathrm{factor}\right)}{\mathrm{Volume}\;\mathrm{of}\;\mathrm{culture}\;\mathrm{well}\;\left(1000\;\mu L\right)}$$

### 4.6. Tissue Protein Extraction and Antimicrobial Peptide Quantification by ELISA

At the end of the infection stimulus (2, 4, 8, or 24 h), the fetal membranes were mechanically separated into the CHD and AMN in order to evaluate the compartmentalized synthesis of AMP. Each layer was placed in independent sterile tubes with protein lysis buffer (20 mM Tris HCl, 150 mM NaCl, 1 mM MgCl_2_, 1 mM EGTA, and 1:1000 P8340 protease inhibitor (Sigma-Aldrich, St Louis, MO, USA)), disrupted by vortex agitation for 10 min at 4 °C, and sonicated at 80 W in 30 s cycles at 4 °C for 15 min in an ultrasonic bath sonicator (Bioruptor^®^ Pico Sonication Device, Diagenode, Denville, NJ, USA). The samples were centrifuged at 3500 rpm for 15 min at 4 °C and the supernatant was collected and stored at −70 °C until the quantification of antimicrobial peptides. After thawing, the protein content of the tissue lysates was quantified by the Bradford method. The samples were adjusted to load 25 µg of AMN total protein per well and 50 µg of CHD total protein per well for the ELISA procedure.

The amounts of HBD-1, HBD-2, HBD-3, and HBD-4 in the tissue lysates were quantified using Peprotech (Minneapolis, MN, USA) ELISA commercial kits (900-K202, 900-K172, 900-K210, and 900-K435, respectively), with an 8 pg/mL detection limit for HBD-1, HBD-2, and HBD4, and a 64 pg/mL detection limit for HBD-3. LL-37 was quantified using a Cloud-Clone Corp ELISA kit (CEC419Hu) (Katy, TX, USA) with a detection range of 123.5–10,000 ng/mL. Elafin was quantified using an Abcam ELISA kit (ab100658) (Waltham, MA, USA) with a detection range of 6.86–5000 pg/mL. RNase-7 (1.56–100 ng/mL), RNase-8 (1–30 ng/mL), sPLA2 (0.312–200 ng/mL), and S100A7 (0.9–60 ng/mL) were quantified using MyBioSource ELISA kits (MBS922852, MBS900616, MBS265046, and MBS936700, respectively) (San Diego, CA, USA). HNP-1-3 were quantified using a CUSABIO Biotech ELISA kit (CSB-E11742h) (Houston, TX, USA) with a detection range of 1.56–100 ng/mL. Assays were performed according to the manufacturer’s instructions. AMP concentrations were normalized per mg of protein.

### 4.7. Cytokine Quantification by ELISA

TNF-α and IL-1β secretion in the culture medium was evaluated by an ELISA commercial kit (DTA00D, R&D Systems, Minneapolis, MN, USA, and 900-K95, Peprotech, Minneapolis, MN, USA, respectively). Both kits had a detection limit of 15 pg/mL. Cytokine levels were normalized per gram of wet tissue.

### 4.8. Staining Techniques

At the end of the experimental procedure, whole chorioamniotic membranes were fixed in 10% formalin for at least 24 h. The tissues were embedded in paraffin blocks and cut into 10 µm slices. The slides were deparaffinized and cleared in Histo-Clear (National Diagnostic, Atlanta, GA, USA) and rehydrated through graded concentrations of ethanol in water for staining. The sections were stained for modified Gram (Remel, Lenexa, KS, USA) [71] or Sirius red/fast green (Sigma-Aldrich, St. Louis, MO, USA) [72]. Gram dye allowed the visualization of Gram-negative bacteria in the tissues as pink- to red-colored bodies. As Sirius red magnifies the usual birefringence of collagen fibers, while fast green counterstains non-collagen proteins in tissue sections, we employed them to evaluate the extracellular matrix integrity in the chorioamniotic membranes.

### 4.9. Microscopy

Images were obtained using a Carl Zeiss Lab A.1 microscope equipped with an AxioCam Erc5s camera (Carl Zeiss, Inc., Thornwood, NY, USA) at 5X magnification. Digital processing of the images was performed with Zen 2.3 (blue edition) software (Jena, Germany).

### 4.10. In Vitro Microbicidal Activity of Recombinant HBDs

In order to know which defensin more efficiently inhibited *E. coli* proliferation, we incubated a constant load of bacteria (1 × 10^5^ CFU/mL) in the presence of 2000 ng/mL of each recombinant, HBD-1, HBD-2, HBD-3, or HBD-4, alone (Peprotech, Minneapolis, MN, USA. 300-51A, 300-49, 300-52, and 300-65, respectively) or combined (500 ng HBD-1 + 500 ng HBD-2 + 500 ng HBD-3 + 500 ng HBD-4). The experimental microbicidal assay was developed in DMEM antibiotic-free medium. As a control for bactericidal activity, a commercial ThermoFisher (Waltham, MA USA) antibiotic mixture was used (15240062): penicillin 100 U/mL, streptomycin 100 µg/mL, and amphotericin B 25 ng/mL. The bacteria were maintained at 37 °C in a humidified chamber with 5% CO_2_. The viable bacteria were estimated by CFU counts in TSA plates from each experimental treatment at time 0 h and 4 h. Each treatment was repeated on at least four different occasions in triplicate.

### 4.11. Statistical Analysis

Descriptive statistics were obtained for each variable. Data normality was evaluated by the Shapiro–Wilk test. Normally distributed data were analyzed by a one-way ANOVA, followed by a Holm-Sidak post-hoc test for multiple comparison analysis; data were graphically presented as the means and standard deviations. Data with no normal distribution were analyzed by a one-way ANOVA, followed by Dunn’s post-hoc test and were graphically presented as boxes (25th, 50th, and 75th percentiles) and whiskers (5th and 95th percentiles). Statistical differences were calculated using a specialized software package (SigmaPlot 11.0, Jandel Scientific) (Palo Alto, CA, USA). Differences were considered statistically significant at *p* < 0.05.

## 5. Conclusions

In summary, our results show the constitutive expression of tissue AMP in chorioamniotic membranes and the ability to increase various AMP after an *E. coli* challenge ex vivo in a temporal- and tissue-specific manner. Particularly, we observed the AMN quickly induced production of HBD-1, HBD-2, and LL-37 and later induced the synthesis of HBD-1, HBD-2, HNP-1-3, S100A7, sPLA2, and elafin. On the other hand, the CHD contributed with a delayed induction of LL-37. Despite the later conclusion, bacteria were able to attach, invade and degrade the tissue, and migrate from the CHD to the AMN in this model, most likely due to protease activity upon the endogenous AMP. These observations are in line with the morphological changes associated with PROM, which deeply increases the risk for PTB.

A highlight of our study was the increased bactericidal activity achieved by the mixture of recombinant HBDs 1-4 at very low concentrations, which encourages further research to study AMP combinations that may offer synergy to control drug-resistant infections in the perinatal period.

## Figures and Tables

**Figure 1 ijms-23-02994-f001:**
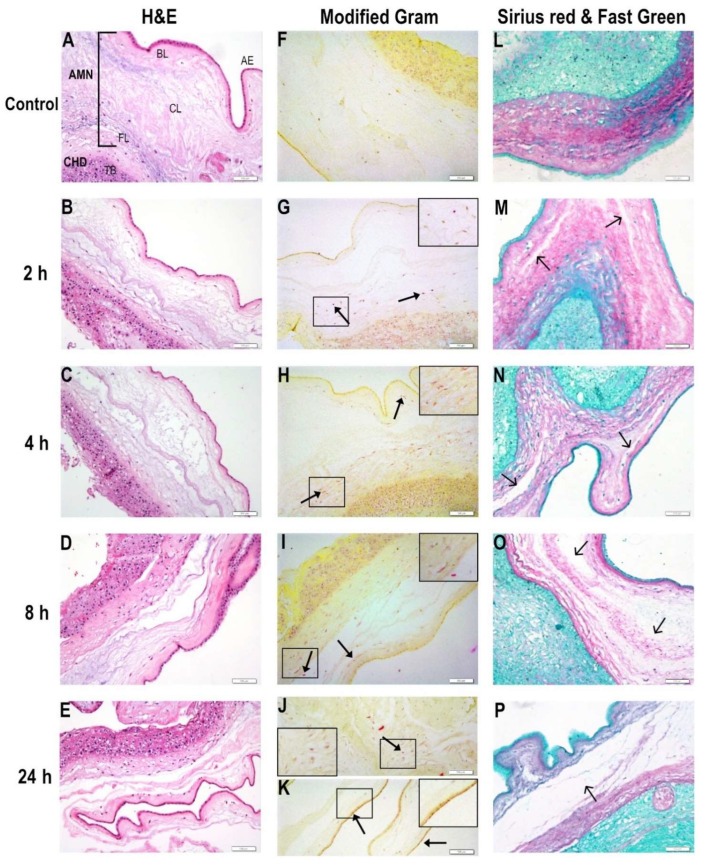
**Kinetic time follow-up of choriodecidual infection by *Escherichia coli*.** The histological examination of *E. coli*-infected membranes was analyzed 2, 4, 8, and 24 h after tissue inoculation to evaluate the mechanical damage and bacteria migratory capacity through the CHD and AMN layers. Left panel (**A**–**E**): Fetal membranes were stained with hematoxylin and eosin. Middle panel (**F**–**K**): Modified Gram staining. The closed arrowheads show bacterial accumulation. Panel amplification is shown in the square. Right panel (**L**–**P**): Collagen fibers were stained with Sirius red to observe zones with collagen degradation. The open arrowheads show zones with extracellular matrix degradation. Scale bar at 100 μm is shown at bottom right in each picture. Representative photographs from five independent experiments are shown. AE: amniotic epithelium. AMN: amnion. BL: Basal layer. CHD: choriodecidua. CL: Compact layer. FL: Fibroblastic layer. TB: Trophoblasts.

**Figure 2 ijms-23-02994-f002:**
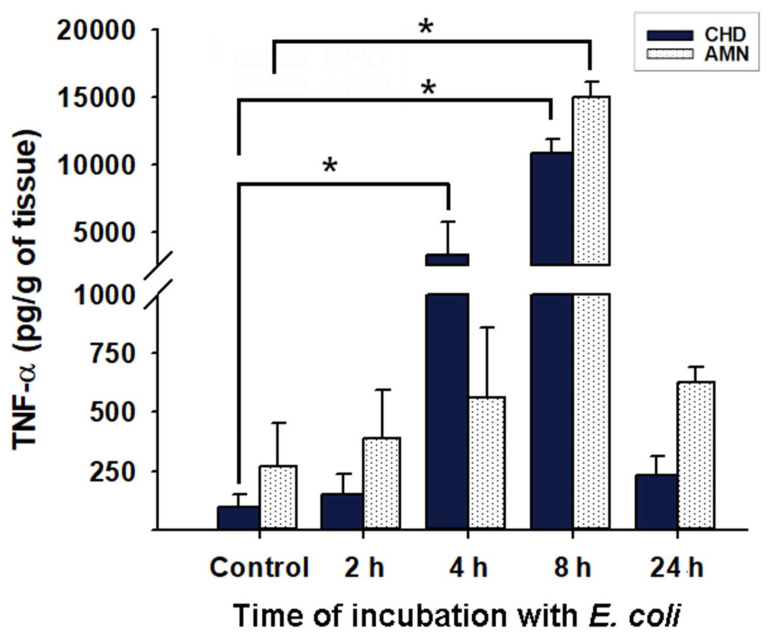
**Pattern of TNF-α secretion into the culture medium from the chorioamniotic membranes infected with *Escherichia coli*.** An inoculum of 1 × 10^5^ CFU/mL *E. coli* was used to infect the CHD compartment. Tissue culture proceeded for 24 h, and an aliquot of the medium was used to assess the TNF-α concentration at each follow-up time point (control, 2, 4, 8, and 24 h). The data are presented as means ± SD. *n* = 3 independent experiments in duplicate. * *p* < 0.05. AMN: Amnion. CHD: Choriodecidua.

**Figure 3 ijms-23-02994-f003:**
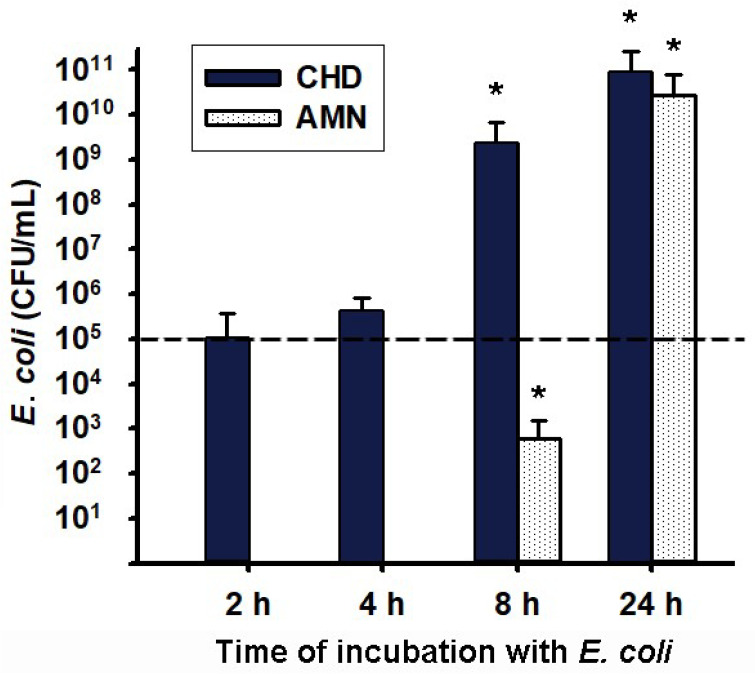
**Transmembrane migration of *Escherichia coli* from choriodecidual to amnion culture media.** An inoculum of 1 × 10^5^ CFU/mL *E. coli* was used to infect the CHD compartment (shown as a dashed line). The tissue culture proceeded for 24 h, and an aliquot of the media was used to assess the CFU/mL at each follow-up time point (2, 4, 8, and 24 h). The data are presented as means ± SD. *n* = 6–7 independent experiments in triplicate. In the case of the CHD (dark bars), * *p* < 0.05 vs. the initial load of bacteria, represented by the dashed line. In the case of the AMN (white bars), * *p* < 0.05 vs. 2 h and 4 h, which were time points with undetectable bacteria, given that the inoculum was placed in the CHD compartment. As depicted, the bacteria migrated towards the AMN and were significantly detected at 8 h and 24 h. AMN: Amnion. CHD: Choriodecidua.

**Figure 4 ijms-23-02994-f004:**
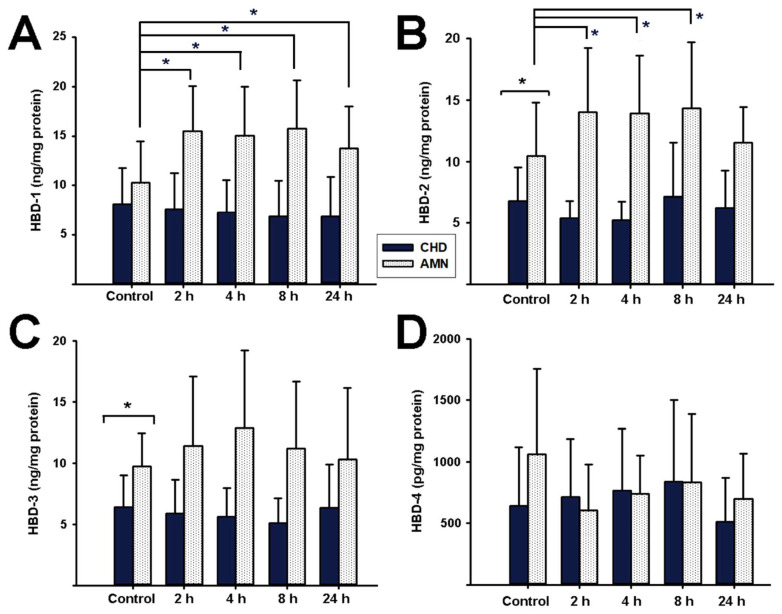
**Effect of choriodecidual infection with *Escherichia coli* upon human beta defensins synthesis.** An inoculum of 1 × 10^5^ CFU/mL *E. coli* was used to infect the CHD compartment. The tissue layers were lysed to assess HBD content at each follow-up time point (control, 2, 4, 8, and 24 h). (**A**) HBD-1, (**B**) HBD-2, (**C**) HBD-3, and (**D**) HBD-4. Beta defensin synthesis by the amnion (AMN) is shown in white bars, whereas synthesis by the choriodecidua (CHD) is shown in dark bars. The data are presented as means ± SD. *n* = 6 independent experiments in duplicate. * *p* < 0.05.

**Figure 5 ijms-23-02994-f005:**
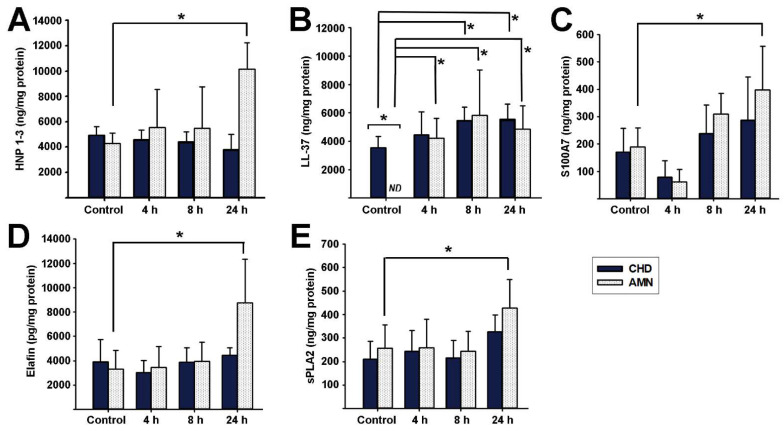
**Effect of choriodecidual infection with *Escherichia coli* upon antimicrobial peptide synthesis.** An inoculum of 1 × 10^5^ CFU/mL *E. coli* was used to infect the CHD compartment. The tissue layers were lysed to evaluate the specified AMP content at each follow-up time point (control, 4, 8, and 24 h). (**A**) HNP-1-3, (**B**) LL-37, (**C**) S100A7, (**D**) Elafin, and (**E**) sPLA2. AMP synthesis by the amnion (AMN) is shown in white bars, whereas synthesis by the choriodecidua (CHD) is shown in blue bars. The data are presented as means ± SD. *n* = 4–5 independent experiments in duplicate. * *p* < 0.05. ND = non-detectable.

**Figure 6 ijms-23-02994-f006:**
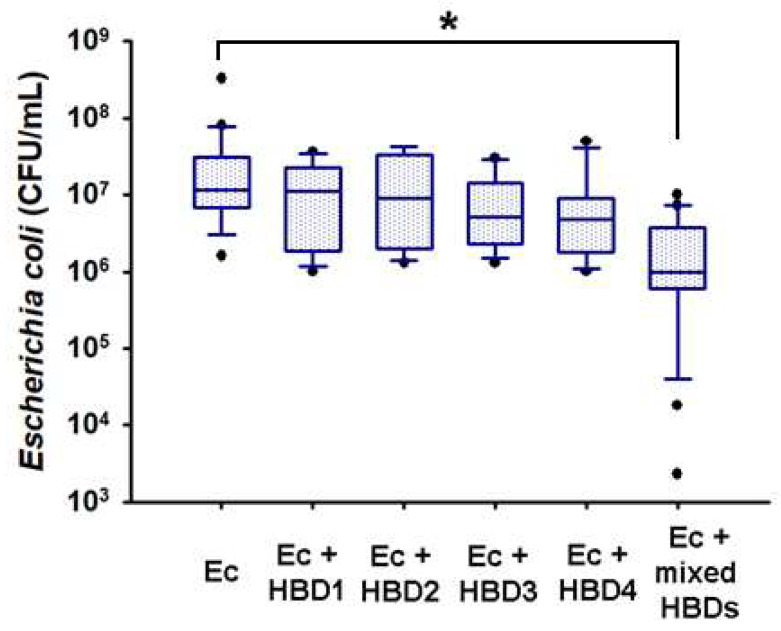
**Microbicidal effect of recombinant HBDs upon *Escherichia coli* growth.** A constant load of bacteria (1 × 10^5^ CFU/mL) was incubated in the presence of 2000 ng/mL of each recombinant defensin alone or combined (500 ng/mL HBD-1 + 500 ng/mL HBD-2 + 500 ng/mL HBD-3 + 500 ng/mL HBD-4) in antibiotic-free medium. Incubations proceeded at 37 °C for 4 h. The *E. coli* control represents the normal bacteria growth in the absence of antimicrobial peptides. The data are presented as boxes (25th, 50th, and 75th percentile) and whiskers (5th and 95th percentile). *n* = 4–8 independent experiments in triplicate. Ec = *Escherichia coli.* * *p* < 0.05.

**Figure 7 ijms-23-02994-f007:**
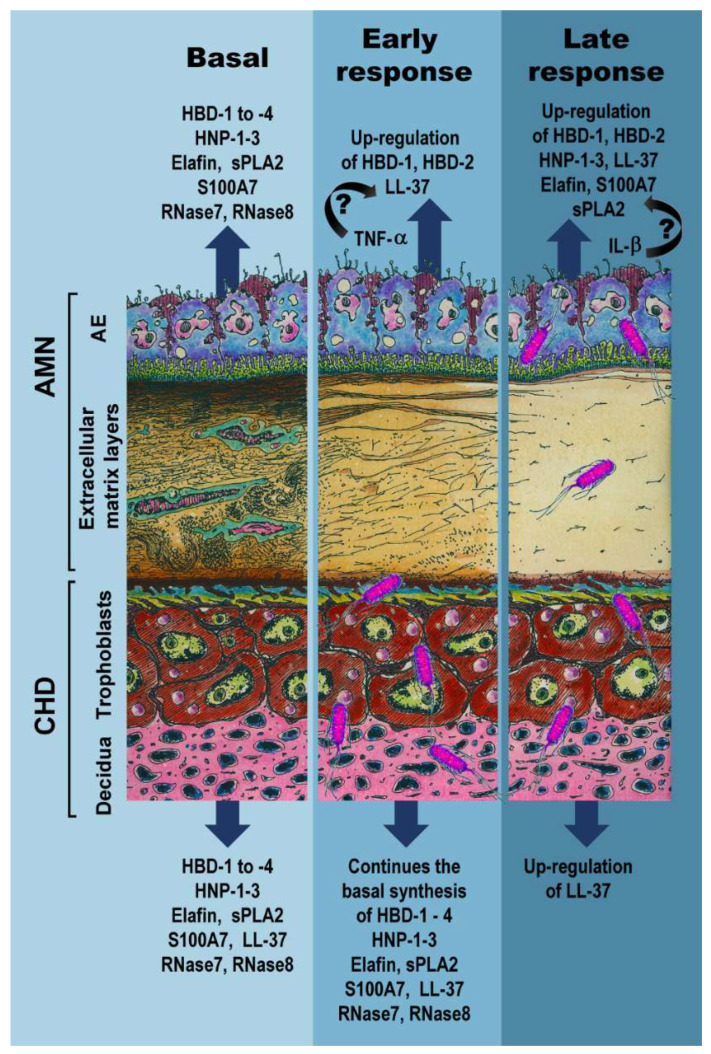
**Schematic overview of AMP synthesis and the mechanical structure of chorioamniotic membranes in response to an *Escherichia coli* choriodecidual infection.** In the basal state, the chorioamniotic membrane presents fully intact fibers of collagen. The AMN and CHD basally synthesize HBD-1 to -4, HNP-1-3, elafin, sPLA2, S100A7, RNAse-7, and RNAse-8, whereas LL-37 is basally absent in AMN. The challenge with a choriodecidual infection by *E. coli* disrupts the mechanical integrity of the collagen fibers and favors a TNF-α inflammatory milieu. The AMN side induces the early (2–4 h) synthesis of HBD-1, HBD-2, and LL-37 in response to *E. coli*. We hypothesize that TNF-α may drive the first AMP wave. At this time point, the CHD continues synthesizing all basally produced peptides. Bacteria bodies are observed only in the CHD compartment. In the late response (after 8 h of infection with *E. coli*)*,* the extracellular matrix of the AMN was practically destroyed, and it appears to be an inflammatory environment enriched by IL-1β instead TNF-α. In this late wave after bacterial challenge, AMN synthesis of HBD-1, HBD-2, HNP-1-3, LL-37, S100A7, elafin, and sPLA2 are induced, whereas LL-37 is upregulated in the CHD compartment. IL-1β may control this observed second AMP wave. By this time, bacteria reach the amniotic compartment. AE: amniotic epithelium. AMN: amnion. CHD: choriodecidua.

**Table 1 ijms-23-02994-t001:** Clinical data of mothers and newborns.

Clinical Parameter	Mean ± SD	Range
	(*n* = 13)	(Min–Max)
Maternal age (years)	30.9 ± 5.8	(22–39)
Gestational age (weeks)	38.6 ± 0.6	(38–40)
Number of pregnancies	2.5 ± 0.7	(2–4)
Newborn cephalic perimeter (cm)	34.8 ± 0.9	(33–36)
Newborn length (cm)	48.6 ± 2.5	(46–52)
Newborn weight (g)	3184.1 ± 453.7	(2760–4040)
Apgar 1 min	8.6 ± 0.4	(8–9)
Apgar 5 min	8.9 ± 0.2	(8–9)
Newborn sex (male/female) (%/%)	5/8	38%/62%

## Data Availability

The data used to support the findings of this study are available from the corresponding author upon request.

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
