# Peer review of "Compartmentalized Innate Immune Response of Human Fetal Membranes against Escherichia coli Choriodecidual Infection"

_ijms, 2022, doi:10.3390/ijms23062994_

Round 1

Reviewer 1 Report

Referee Report - ijms-1583772

This is a well-written paper regarding the production of antimicrobial peptides in response to an ex vivo infection of the choriodecidua by E. coli. The paper is informative and deserves to be published. However, I think that there are a number of points that need to be addressed to improve the paper and make it ready for publication (see below).

Page 2, line 82: HNP are also often referred to as human neutrophil defensins and this should be added here.

Page 2, lines 90-92: This sentence requires a reference.

Page 3, line 105: ‘within’ not ‘upon’.

Page 5, line 150: ‘below’ not ‘beyond’.

Page 5, lines 149 – 160: Can the authors offer an explanation why they think that the concentration of TNFα decreased at the 24 h time-point?

Figures 2-6: The authors use * to indicate statistically significant results. However, it is not clear in all cases which 2 results they are distinguishing between. The addition of a bar linking the 2 results where significance is derived from would make the data easier to interpret and enhance the manuscript.

Section 2.4: Whilst in many tissues HBD1 is constitutively produced, many other AMP are only normally induced upon insult, e.g. infection. Can the authors explain why detectable levels of HBD-2, HBD-3, HBD-4, etc were present in control samples? Is it possible that the processing steps to prepare the membranes for experiments have caused a physical ‘insult’ and this has caused induction of the synthesis that are normally only induced, or is this something that is specific to the amnion and choriodecidua?

Section 2.5 & Figure 6: Can the authors provide a rationale why these concentrations of HBDs were selected to assay for antimicrobial activity? Were any other concentrations tested?

Section 2.5: As these HBD (and combinations) are relatively ineffective (maximum ~1 log reduction in cfu), it would be useful for the authors to indicate that the primary role of these AMP might be in their ability to recruit cells to the site of infection and that they may be the primary antimicrobial effectors. Unfortunately, the model used cannot demonstrate this.

Page 9, lines 287-291: It would be useful if the authors could link the production of TNFα and IL-1β to AMP production. Can either of these molecules induce, or inhibit, production of AMP?

Page 9, lines 284-286: It seems unlikely that in this model AMP contribute significantly to direct antimicrobial activity, given the relatively poor antimicrobial activity described in Figure 6. Therefore, it would be useful for the authors to describe potential other roles for AMP in the defence against infection, e.g. recruitment of immune cells.

Figure 7 legend: What evidence do the authors have that IL-1β controls the observed second AMP wave? Measurements of AMP, IL-1β and TNFα between 8 and 24 h may have provided some evidence for this, but the current data does not support this conclusion and this should be made clear in the current manuscript.

Page 11, lines 330 – 344: The antimicrobial activity of the combination of HBDs in Figure 6 is relatively minor (~ 1 log kill). This is neither ‘remarkable’ (line 330) or ‘highly effective’. The description of the antimicrobial activity observed should reflect this better and this paragraph should be re-worded.

Page 11; lines 336 – 338: The different targets and different mechanisms of action referred to in this section should be explained in more detail, with suitable references from the literature.

Section 4.1: Details of ethical approval should be included in this section.

Page 13, line 427: The ‘2’ in MgCl2 should be subscript.

Section 4.5: Why were the concentrations of samples adjusted to load 25 µg of amnion and 50 µg of choriodecidua, and not the same concentration of each?

Sections 4.5 & 4.6: Why were the AMP samples adjusted per mg of protein, whereas the concentrations of IL-1β and TNFα were normalized per gram of wet tissue?

Page 14; line 498: I do not think that antimicrobial activity was ‘strong’ and this language should be tempered. How did the authors distinguish between bacteriostatic and bactericidal activity? This is not clear from either the results or methods sections, and the methods section contains details of only how to determine bactericidal activity.

Reviewer 2 Report

The manuscript studies the compartmentalized innate immune response of fetal membranes during E. coli infection. The authors describe the gradual migration of E. coli from the choriodecidual region to the amnion by monitoring for 24 hours and using their own previously published methodology. The presence of antimicrobial peptides (AMPs) at each stage of infection is evaluated and the possibility of remission of infection by the use of human beta-defensins is studied. Progressive bacterial colonization accompanied by tissue destruction and inflammation as measured by the presence of TNF-alpha and IL-1beta is demonstrated. The secretion of AMPs is identified as occurring in two distinct phases (early response and late response). Interestingly, the microbicidal use in vitro of several defensins results in a synergy.

Comments: The evaluation using culture of chorioamniotic membranes is appropriate, original and brings us closer to understanding the pathogenesis of this type of infection.

The microbiological techniques used for the evaluation of bacterial infection and the detection of cytokines or AMPs are appropriate and are clearly described in Material and Methods.

Interesting findings are: 1) the observation of bacterial migration associated with tissue destruction and inflammation, 2) the different pattern of AMPs secreted according to the phase of response, 3) the synergy observed in vitro of different beta-defensins in spite of using sublethal concentrations (nM), a finding that could have a future therapeutic application against this type of infection especially if caused by antibiotic-resistant E. coli.

The manuscript is well organized, well written and understandable. The bibliography is adequate, current and complete.

Main question:

The authors should note that the in vitro antimicrobial activity of some human defensins decreases in the presence of salt (>50 mM NaCl). If each of the peptides were solubilized in a salt-free buffer or medium other than the medium used (i.e. DMEM) and then added to the bacterial suspension it is possible that the higher antimicrobial activity of the 4 defensins together may be due to a dilution effect (lower salt concentration). Please indicate in which medium the defensins were dissolved to avoid this speculation.

Typographical corrections:

I suggest a revision of the microbial nomenclature used in some parts of the main text (e.g., line 241 E. coli) or in the References section (lines 546, 607, 615, 624, 628, 649, 655, 658, 659) due to genus and species names should be written in italics and in lower case the name of species (e.g. E. coli).

Reviewer 3 Report

This is an interesting and significant paper reporting novel results of the ongoing research of the research group. Using an in vitro model of choriodecidual infection resulting from ascending E.coli colonization, they demonstrate increasing expression of tissue AMPs in chorioamniotic membranes in a temporal- and tissue-specific manner.

However, the microbiological part of the experiments shows some weaknesses and makes it difficult to accept the in vitro model representing an ascending infection. In my opinion, some additional information and corrections could further improve the quality of the manuscript.

Major concerns:

4.3., Lines 394-398:  The characterization of the strain E.coli used in the experiments is inadequate and insufficient. The authors cite a reference, where PCR-based virulence factor-genotype characterization and grouping of E. coli has not even been described. Please provide detailed information regarding serotyping (O, H K) and virulence factors tested in the multiplex PCR. B2 phylogroup of E.coli can be considered as pathogenic, but not as uropathogenic. Furthermore, the clinical situation (isolation of the strain from a neonate with clinical sepsis) only support the invasive character of the bacterium but not proves prenatal ascending urogenital infection of the mother. The authors should also explain why they didn’t choose a well characterized ATCC reference strain of E.coli (e.g. Escherichia coli O6:H1 (strain CFT073 / ATCC 700928 / UPEC). In Fig. 7, the authors draw a flagellated E.coli without any information, whether the strain used in the experiments had H antigens.

4.3. Lines 409-411: 1x105 CFU/ml is a typical bacterial concentration found in the urine in the case of a manifest urinary tract infection but cannot be reached in early phases of an infection. In my opinion, during these experiments, the chorioamniotic membrane culture was simply overflowed with bacteria, a situation which is not considered as physiological. Are there any data from the literature supporting the use of this high bacterial concentration in similar in vitro models?

            Minor comments (please include the answers also in the manuscript):

  1. Please provide data regarding the bacterial etiology of chorioamnionitis and explain why you choose E. coli for testing.
  2. Are there any other possible antimicrobial effects of AMPs beside bacteriostatic/bactericide activity?
  3. Which cells are the source of AMPs and cytokines like TNF-alfa and IL—beta in the in vitro model?
